# Fractionation of Anaerobic Digestion Liquid Effluents through Mechanical Treatment and Filtration

Themistoklis Sfetsas [1,*], Georgia Sarikaki [1], Afroditi G. Chioti [1], Vassilis Tziakas [1], Polycarpos Falaras [2] and George Em. Romanos [2]

1   QLAB Private Company, Research & Development, Quality Control and Testing Services, 57008 Thessaloniki, Greece; g.sarikaki@q-lab.gr (G.S.); chioti.a@q-lab.gr (A.G.C.); billtzi@ergoplanning.gr (V.T.)
2   Institute of Nanoscience and Nanotechnology, National Center of Scientific Research "Demokritos", Agia Paraskevi, 15310 Athens, Greece; p.falaras@inn.demokritos.gr (P.F.); g.romanos@inn.demokritos.gr (G.E.R.)
*   Correspondence: t.sfetsas@q-lab.gr; Tel.: +30-2310-784712

**Abstract:** Organic wastes and by-products from several activities, including food industries, farming, and animal husbandry, are a hygiene threat when aerobically decomposed. Therefore, their management is crucial for public health. In this direction, anaerobic digestion (AD) systems may be the solution by transforming waste into energy, which may decrease the environmental impact. However, their efficacy should be carefully examined. In this innovative study, we evaluated the physicochemical and microbial characteristics of liquid digestate (LD) retrieved from organic animal wastes in northern Greece using nanofiltration. Using treatment technologies, including physical (solid–liquid separation, microfiltration, and nanofiltration) and biological (anaerobic digestion), heavy metals and microbial (i.e., *Salmonella* spp., *Escherichia coli*, and *Enterococcus faecalis*) concentrations were reduced and nutrients were recovered. This work sets the basis for the efficient management of liquid digestate. Our method may enable the use of treated liquid digestate for unlimited irrigation water and other industrial applications of water. Apart from the sanitation process, the recovery of nutrients for soil fertilization seems to be a more sustainable way for future agricultural practices.

**Keywords:** anaerobic digestion; liquid digestate; solid fibrous digestate; fractionation; microfiltration; nanofiltration



## 1. Introduction

Poor management of organic wastes is a major cause of pollution. Indeed, organic by-products from food industries, farming, and animal husbandry may pose a hygiene threat when aerobically decomposed [1,2]. However, such wastes can be appropriately managed using anaerobic digestion (AD) systems to transform waste into energy, which may decrease the environmental impact and increase reusability [1]. Specifically, AD refers to the biological digestion of organic matter under anaerobic conditions, occurring in aquatic environments and involving different microorganisms, wherein a diverse community of microorganisms converts complex organic matter into biogas and whole digestate (WD) [3]. Although AD is a highly favorable waste treatment technology, particularly from an environmental standpoint, it cannot achieve complete waste stabilization [4].

During the digestion of various animal wastes, particularly those of cattle, pig, poultry, and sheep manure, pathogenic microbes (*Escherichia coli*, *Salmonella* spp., *Listeria monocytogenes*, *Clostridium perfringens*, *Campylobacter jejuni*, *Cryptosporidium parvum*, *Giardia intestinalis*, and *Clostridium botulinum*) are able to survive the digestive process and remain in the digestate [5]. Consequently, it is of utmost importance to apply proper disinfection processes to avoid pathogens' transportation from agricultural land through the food chain to humans.

Whole digestate (WD) sanitation is based on several factors, such as the quality of substrates fed into the reactor, reactor performance, digestion temperature, slurry retention time, pH, and $NH_3$ concentration [6]. Methods, including pasteurization, chlorine treatment, UV-light exposure, ozone treatment [7], and high-pressure treatment within a vessel [8], can be performed to reduce the pathogen load in the final WD effluent.

Alternative methods (i.e., electro-technology, microwave treatment, pressurization, and ultrasound treatment) have been developed and performed to reduce bacterial populations. In 2018, Uggetti et al. presented a sustainable approach to managing wastewater, which is considered a process for resource recovery waste treatment [9]. During this research, an experimental microalgae-based wastewater treatment system was developed using three semi-closed horizontal photobioreactors under the European project INCOVER to reuse it and produce added-value products. There were low energy requirements for growing microalgae, using agricultural and sewage wastewater as feedstock. Their findings were very encouraging, as biomass production reached almost 2.2 kg VSS/d with compensatory wastewater treatment performances (<2 mg/L for phosphates, <10 mg/L for ammonia, and <15 mg/L for nitrites and nitrates) [9].

Management of the digestion residue, if intended as either a soil improver or organic fertilizer, should comply with Circular Letter 3891/134991, 1-12-2016, "Management of livestock and slaughterhouses manure, and digestion residue from biogas plants" and Regulation (EC) No 1069/2009 in regard to animal by-products. Application of the digestion residue from a biogas plant as a fertilizer or soil conditioner requires the application of various sanitary precautions depending on the type and risk category of the animal raw material used. According to the European Union (EU) Regulation No 142/2011, re samples of decomposition residues must comply with limits set by the regulation regarding the microorganisms *Escherichia coli*, *Enterococcaceae*, and *Salmonella* [10]. Moreover, Regulation (EU) 2019/1009 introduces harmonisation rules for compost and digestates as components of fertilizers in the EU [11].

However, the corresponding circular letter offers suggestions for alternative sanitation methods, while pasteurization is a non-effective practice regarding economic and energetic aspects. Therefore, novel alternative strategies for microbial load reduction are developed so that the decomposition residue utilized in the fields does not constitute a risk to public health. In fact, Circular Letter 969/14986/21-6-2019 of the Department of the Directorate of Health and Safety of the Hellenic Ministry of Rural Development and Food defines the critical parameters that must be proven to be reduced in order to develop an approved method for sanitizing digestate [12].

Although several works have been performed on a laboratory scale for the treatment of wastes using green technologies (e.g., using filters filled with recovery materials or pretreating digestates with cetyltrimethylammonium bromide and coal fly ash) with sufficient performance and encouraging results, to our knowledge, few studies have been conducted in real-world settings [13,14]. Therefore, to fulfill this knowledge gap, we developed a pilot sanitation unit using a subsequent filtration with decreasing porosity in liquid digestate to achieve a clear, purified liquid, which would then be used for unlimited irrigation and other industrial uses of recirculated cooling water according to Hellenic Joint Ministerial Decision 145116/02-02-2011 (Table 1) [15].

**Table 1.** Limits for microbiological and conventional parameters, as well as the minimum required treatment for reuse of treated liquid waste, as per the Hellenic Joint Ministerial Decision 145116/02-02-2011 (Official Government Gazette B 354/2011) [15].

| Type of Re-Use | *Escherichia coli* (cfu/100 mL) | BOD5 (mg/L) | Suspended Solids (SS) (mg/L) | Turbidity (NTU) | Minimal Treatment | Total N (mg/L) | Total P (mg/L) |
|---|---|---|---|---|---|---|---|
| Limited irrigation and disposable cooling water | ≤200 median value | ≤25 | - | - | Secondary with disinfection | <45 | - |
| Unlimited irrigation and recirculated cooling water (boilers and processes) | ≤5 for 80% of samples and ≤50 for 95% of samples | ≤10 for 80% of samples | ≤10 for 80% of samples | ≤2 median value | Secondary and tertiary with disinfection | <15 | <2 |
| Urban use and enrichment of underground aquifers and peri-urban green areas (groves and forests) | Total coliforms ≤2 for 80% and ≤20 for 95% of samples | ≤10 for 80% of samples | ≤2 for 80% of samples | ≤2 median value | Secondary and advanced with disinfection | <15 | <2 |

## 2. Materials and Methods

### 2.1. Equipment for WD Treatment

WD was utilized after AD in a biogas plant (Biogas Lagada S.A., Thessaloniki, Greece) of a 1 MW electrical production capacity. The plant operates with two anaerobic digesters (D1 and D2, 4000 m$^3$ each) connected in series. D1 is fed hourly by an underground 550 m$^3$ tank (liquid feedstock) and a solid feeder (moving floor unit) for solid biomass. After the digestion, the effluents are stored in two storage tanks (ST1 and ST2, 8000 m$^3$ each) and applied further as soil improver in nearby fields.

From D2 the WD can be separated using a mechanical separator CRI–MAN, SM300/75 Pro (CRI–MAN S.p.A., Correggio, Italy) in primary solid and liquid fractions. Then it was further separated with a centrifugal separator Alfa Laval Aldec 45 (Alfa Laval AB, Lund, Sweden) in secondary solid and liquid fractions. With the help of a pumping system consisting of three progressive cavity pumps from Roto Pump Ltd. Noida, India (models: RMC 542 ×2 and RLCB 571 ×1) and a direct drive plunger pump (Model: 2SF05SEEL from CAT PUMPS, Minneapolis, MN, USA), the fluid was led to all filtration units and intermediate tanks. Filtration units were acquired from Pentek, Coraopolis, PA, USA (bag filter), and microfiltration (MF) and nanofiltration (NF) units were acquired from Atech Innovations GmbH, (Gladbeck, Germany).

### 2.2. Fractionation and Filtration Procedure of Digestate

2.2.1. First Treatment Stage

During the first stage of the pilot scale treatment, the initial whole digested residue (WD), which contains 5.0–7.5% solids depending on the seasonal feedstock, is introduced to the mechanical separator (screw press) at a flow rate of 30 m$^3$/h from the digestion tank of the biogas plant. Then, it is separated into the solid fibrous digested residue (SFD$_{sep}$) which now contains 23.0–30.0% of solids, and into the liquid digested residue (LD$_{sep}$) which consists of the remaining 2.5–5.0% of the original solids. The solid fibrous digestate (SFD$_{sep}$) returns to the digestion tank to repeat the process of AD or is stored in a stockpile for field application as a solid organic soil improver, while the liquid digestate (LD$_{sep}$) is transferred to a 20-m$^3$ tank for further treatment. Additionally, the solids that will settle in the tank will again return through a secondary piping circuit to the separator so they can be further separated.

2.2.2. Second Treatment Stage

In the second stage of the treatment process, the liquid digested residue (LD$_{sep}$) is channeled via the 20-m$^3$ tank to the centrifugal separator (decanter) at a speed of 5 m$^3$/h using an industrial progressive cavity pump appropriate for sludge and sewage transport. At this point, the wastewater that has already been treated undergoes more thorough separation. Specifically, the centrifuge (decanter) separates the liquid digested residue that

contains 2.5–5.0% solids into a new solid fibrous digested residue ($SFD_{dec}$) that contains 18.0–23.0% of the solids and a liquid digested residue ($LD_{dec}$) that has the remaining 1.2–2.5% of solids [7]. The newly separated (secondary) solid residue ($SFD_{dec}$), with the use of a screw conveyor, ends up in a storage tank of 5 m³; while the newly separated liquid residue is transferred with a progressive cavity pump at a speed of 3 m³/h to a 6 m³ tank, that serves as an intermediate tank for the following stage. At this point, the outgoing liquid residue ($LD_{dec}$) will be recirculated from a secondary piping circuit in order to dilute the incoming wastewater ($LD_{sep}$). This procedure aims to minimize the number of solids in the outgoing wastewater.

### 2.2.3. Third Treatment Stage

Then, the third stage of the treatment process follows, which is referred to as filtering and MF, making effluent suitable for the fourth and final stage (Figure 1). During this course, a progressive cavity pump initially moves the liquid fraction at a 2–3 m³/h flow rate through a polypropylene bag filter, which retains solids > 200 μm that do not settle as sediment in the centrifuge. The liquid fraction then passes through the MF assembly, allowing only solids ≤ 1.2 μm to pass through.

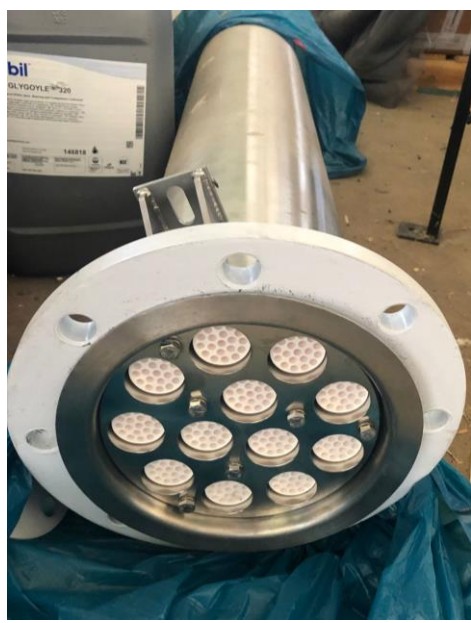

**Figure 1.** Microfiltration unit.

Then the filtrate exits the MF system at a flow rate of 0.2 m³/h and ends up in a 1-m³ storage tank, where sensors check it for electrical conductivity (EC) and turbidity.

### 2.2.4. Fourth Treatment Stage

In the fourth and last stage, a triple piston pump draws the sample from the 1-m³ tank and pushes it to the nanofiltration unit with a flow rate of 50 L/h. This unit is able to remove the remnants of organic pollutants and pathogenic organisms from wastewater through nanofiltration.

After the last NF stage, the water is collected in a 300-L tank, in which EC and turbidity measurements are made on an automated basis.

As mentioned earlier, a central computer with Lab View is responsible for the process. At the same time, it monitors the values before and after the NF unit, checks the system's flow and peripheral pressure, and determines centrifugation data from a Decanter.

Following the above, the pilot plant had the following mechanical systems and equipment requirements (Figures 2 and 3):

- A separator (screw press);
- Centrifuge (Decanter);
- Three pumps;
- A microfiltration system;
- Six tanks of different sizes;
- A screw conveyor for the removal of solid digested residue;
- One bag filter;
- A nanofiltration unit with its support scaffold;
- An electrical panel of dimensions $1.0 \times 1.0$ m;
- An air conditioner for temperature reduction inside the container;
- Two ventilation outlets;
- And a host computer for entire pilot plant management.

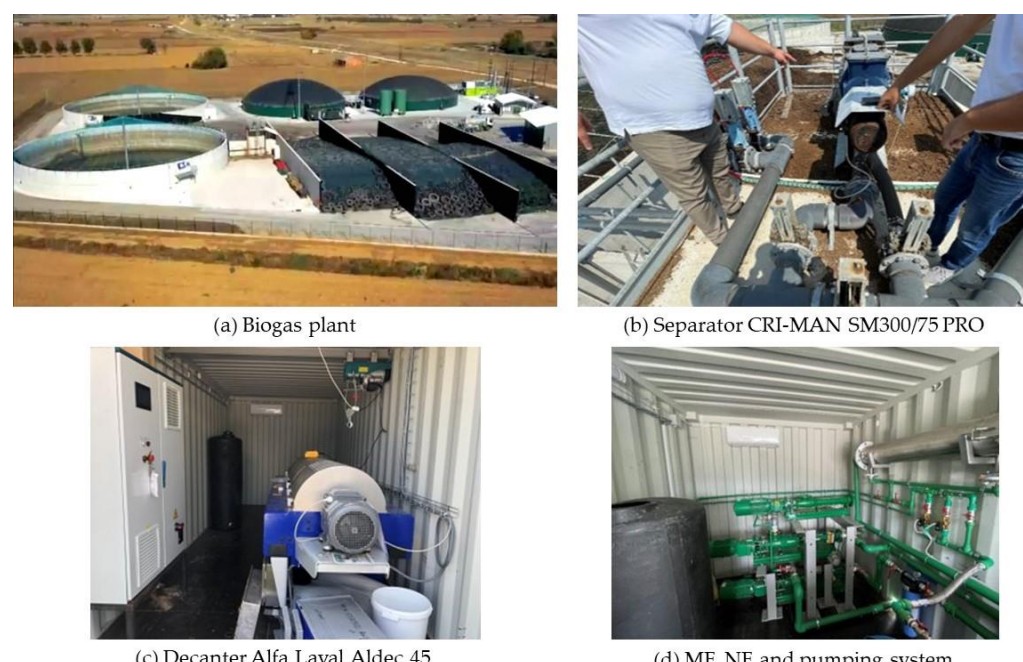

(a) Biogas plant　　　　　　　　　　(b) Separator CRI-MAN SM300/75 PRO

(c) Decanter Alfa Laval Aldec 45　　　　　　　　(d) MF, NF and pumping system

**Figure 2.** Photographs of (**a**) biogas plant; (**b**) mechanical separator; (**c**) centrifugal separator; and (**d**) microfiltration and nanofiltration units.

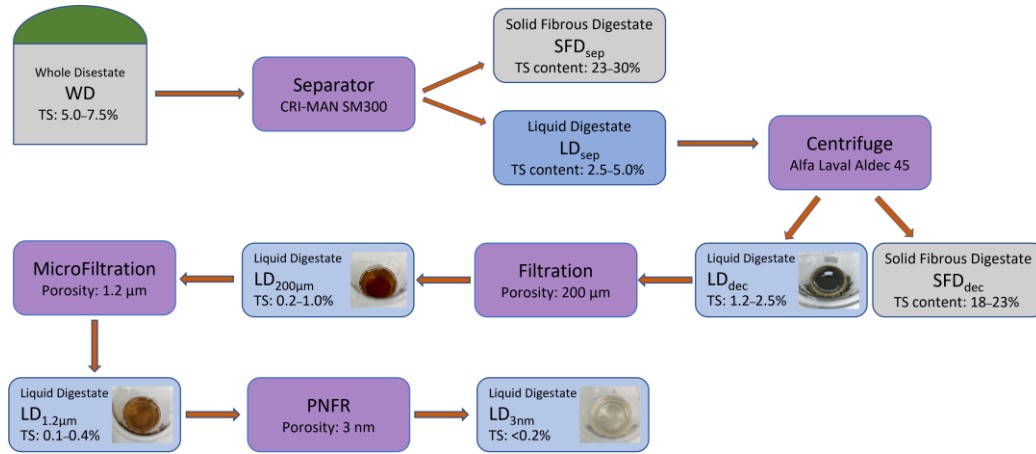

**Figure 3.** Digestate treatment flowchart according to the project Photosan.

Apart from the screw conveyor and the tanks, everything else was placed inside a container.

### 2.3. Physicochemical Methods

The determination of values on the pH scale was carried out with the method APHA 4500-H$^+$ using a HACH instrument (HACH Model HQ30D; HACH, Loveland, CO, USA) equipped with a universal pH measuring electrode (924 001) and a temperature measuring electrode (027 500) [16].

The determination of EC was based on EN 13038 Standard–Determination of EC. The sample was initially diluted with deionized water and then was measured using an HQ30D digital multimeter kit and conductivity electrode (HACH Model HQ30D).

For the determination of total solids (TS), total suspended solids (TSS), and total dissolved solids (TDS), samples were dried at $103 \pm 2$ °C and $180 \pm 2$ °C, respectively, employing APHA method 2540-B and Total Suspended Solids APHA 2540-D [17,18]. For the determination of volatile solids (VS), which represents the portion of suspended or dissolved solids lost from a sample upon ignition at a specified temperature for a specified duration, the APHA 2540-E was used [17].

An Agilent 7850 ICP-MS (Agilent Technologies, Santa Clara, CA, USA) equipped with the ORS$^4$ collision cell was used to analyze macro-elements and trace metals. Sampling was performed using an Agilent SPS 4 autosampler. The 7850 ICP-MS was configured with the standard ISIS 3 injection system. The samples were prepared for analysis according to the digestion procedure outlined in ISO 17294 Part I and II and APHA 3125 [19–21].

The sample is decomposed in acid at a high digestion vessel pressure with the help of a Milestone Ethos Up microwave oven (Milestone Srl, Sorisole, Italy) and the resulting solution is analyzed. First, an amount of sample (0.5–1.0 g) was weighed and $HNO_3$ and $H_2O_2$ were added to the sample followed by gradual digestion up to 210 °C. Then, the sample was diluted and analyzed by ICP-MS.

Chemical Oxygen Demand (COD) is an indicative measure of the oxygen required for the chemical oxidation of organic compounds by a specific oxidant (dichromate ion, $Cr_2O_7^{2-}$) under controlled conditions and is expressed in mass of oxygen consumed. In our work, it was held by using a commercial spectrometer HACH DR 3900, as described elsewhere, based on the Closed Reflux, Colorimetric Method APHA 5220-D [22–24].

Total phosphorus was measured using the molybdovanadate method, HACH reagents, and the HACH DR3900 spectrophotometer. The determination of nitrate–nitrogen concentration is based on the APHA 4500-$NO_3^-$-Ultraviolet Spectrophotometric Screening Method and measurements were conducted at 220 nm with a JASCO V-630 Spectrophotometer [25]. Ammonium–nitrogen concentration was determined photometrically with a JASCO V-630 spectrophotometer at 420 nm according to the APHA 4500-$NH_3$ B and C method [26]. Turbidity was analyzed with a UV-Vis spectrophotometer, COD3 Plus Colorimeter (LaMotte, Chestertown, MD, USA), according to the APHA 2540-E method [17].

### 2.4. Microbiological Methods

#### 2.4.1. Detection of Bacteria *Salmonella* spp.

The method for detection of *Salmonella* spp. was based on ISO 6579-1:2017. Colony-forming microorganisms on solid selective substrates, when tested according to the protocol, demonstrate defined biochemical and serological characteristics. Initially, the sample is pre-enriched in Buffered Peptone Water, at ambient temperature and incubation at 34–38 °C for 18 h. Subsequently, the culture obtained from the first stage was inoculated in two selective substrates and the resulting cultures were recovered and coated in two solid selective substrates, in Xylose Lysine Deoxycholate agar (XLD agar) and in a supplement to XLD agar. Finally, the colonies of potential *Salmonella* were subcultured in a non-selective substrate (nutrient agar), and their identity was confirmed by means of appropriate biochemical and serological tests.

### 2.4.2. Enumeration of Bacteria *Enterococcus faecalis*

Enumeration of *Enterococcus faecalis* is based on a combination of ISO 7899-2:2000 (detection and enumeration of Enterococci in water) and CEN-TR 16193:2013 (detection and quantification of *Escherichia coli* in sewage sludge, treated biowaste, and soil).

The initial dilution (Dilution A) was prepared by weighing 10 g (wet weight) and adding an appropriate amount of peptone saline solution up to a final volume of 100 mL. Then, the material was mixed in the homogenizer for 90 s, aliquoted into containers and centrifuged (1600 rpm, 3 min, $10 \pm 1\,°C$). The supernatant (1 mL) was aseptically vacuum filtered through a 0.45 μm Whatman membrane and the membrane was placed in an SB plate. The plates were incubated inverted at $36 \pm 2\,°C$ for $44 \pm 4$ h. The decimal dilutions of the supernatant were filtered accordingly. After incubation, in the case of typical colony (brown–red color) development, the membrane is transferred to a bile aesculin azide agar medium (preheated to $44 \pm 0.5\,°C$ as a confirmatory step). Black color development on bile aesculin azide agar after 2 h indicates the bacterial growth of *E. faecalis*.

### 2.4.3. Enumeration of Bacteria *Escherichia coli*

Method of detection and enumeration of *Escherichia coli* in the digester material is based on the CEN-TR 16193 (2013) standard. The method is based on a membrane filtration process for quantitative detection, by culturing the individual colonies in a chromogenic culture medium. The method is suitable for estimating the logarithmic reduction of *Escherichia coli* in sanitation processes.

## 3. Results and Discussion

In this study, we have designed and tested a pilot sanitation unit utilizing centrifugation and subsequent steps of filtration with decreasing porosity (micro- and nanofiltration).

Several previous studies have focused on the treatment of digestates. Specifically, a recent study by Shi et al. indicated a novel electrodialysis system development utilizing in situ the anode electrodialysis for the electrochemical oxidation and effective removal of antibiotics in the course of nutrient recovery from pig manure digestate [27]. At the same time, electrochemical oxidation had no significant effect on the nutrient recovery efficiency, but the pathogenic microorganism indicators were efficiently inactivated in the first 30 min. Though by this process a high concentration of disinfection by-products was generated, they were absorbed by anode electrodialysis, resulting in wastewater of very low tri-halomethanes and haloacetic acid concentrations [27].

Moreover, in the study of Maynaud et al., it was reported that pathogenic bacteria in digestates can be inactivated by competition with indigenous bacteria. In the view of the PRObiotic project, the activity of digestate microorganisms, which is related to competition for available nutrients and how it influences the inactivation of pathogenic bacteria, was investigated. Based on the findings, when the availability of organic material and microbial activity increases, *Salmonella enterica* serotype Derby's survival in digestates decreases. Generally, the results of this study demonstrate how understanding the biotic processes involved can help improve microbial control dynamics and microbiological risk management [28].

For the rapid removal of nutrients and ecological inactivation of the pathogens *Clostridium* spp. and *Arcobacter* spp. in swine wastewater, the co-culture of vetiver and *Dictyosphaerium* sp. has been developed by the scientific team of Xinjie et al. Regarding their results, on the 15th day of the culture period, the bacterial community shifted from pathogen-dominant to photobacterial-dominant in the original wastewater. Furthermore, the plant–algae co-culture has decreased the levels of $NH_4^+$—N (from 102 mg $L^{-1}$ to 5 mg $L^{-1}$) and phosphorus below acceptable limits as well as significantly reduced salinity and in-activated pathogens at wastewater treatment facilities within 15 days. The plant–algae co-culture also showed further significant interactions between microalgae and plants, such as water acidification via plant root respiration, algal growth with lower ammonia

toxicity, and bicarbonate stress mitigation by microalgae and plant growth with reduced hypoxic stress, among others [29].

Koziel et al. conducted a treatment of infectious animal carcass digestate utilizing ammonia. Regarding the results, the minimum inhibitory concentration of NH3 was 0.1 M (~1.468 $NH_3$—N mg/L), and 0.5 M $NH_3$ (~7.340 $NH_3$—N mg/L) for ST4232 and MRSA43300 bacterial strains, respectively, at 24 h and pH = 9 $\pm$ 0.1. Furthermore, the increase in NH3 concentration and/or time of treatment increased bacteria inactivation. Despite the complexity of the chemistry and microbiology of the digestate, the treatment with NH3 was effective and consistent using the minimum inhibitory concentration determined in sterile saline solution, except for ST4232 in the late-phase ability of aerobic digestion scenario where the minimum inhibitory concentration was five times greater. Nevertheless, within 24 h, both pathogens were completely inactivated [30].

Moreover, a recent review by Singh et al. highlighted the effect of shear rate on different stages of an anaerobic digestion process and reported that the methane content may vary with the variation in mixing speeds. In addition, the authors stated that the mixing effect is significant in cases where the total solid content is higher and revealed that intermittent mixing is favorable in comparison to continuous mixing [31].

Limited irrigation (no sprinkler application) refers to areas where public access is not expected, such as forage crops, industrial crops, pastures, and non-fruit trees, with the premise that the fruits are not in contact with the soil, seed crops, and product crops, which undergo further processing before consumption. Disposable cooling water for industrial use refers to the supply of underground aquifers by infiltration of an intermediate soil layer with sufficient thickness and suitable characteristics. Unlimited irrigation applies to all crops whose products are consumed raw. In unlimited irrigation, sprinkler application is allowed. Urban use involves watering large urban areas without sprinkler application, extinguishing fires, use for decorative fountains, and street cleaning (Table 1).

For this purpose, we conducted and presented the Continuing Professional Development (CPD) study that concerns the preliminary unit design as well as the preparation of the overall process flow diagram. The goal of CPD studies is to establish the mass and energy balance equations, incorporate Key Performance Indicators (KPIs) of the process (system productivity per time and volume, pressure drop, liquid speed, water perviousness, and solute rejection efficiency), and apply the mathematical description of the filtration process and other cutting-edge technologies (reference cases). With the help of the created CPD tools, it will be possible to evaluate the Photosan pilot unit for the elimination of several organic and inorganic pollutants, bacteria, and pathogenic organisms in typical applications.

The process of wastewater filtration through the membranes, which allows the retention of solids that are larger than the diameter of the membrane's pores, is called Tangential Flow Filtration (TFF). TFF is a filtration method in which the feed flow runs tangentially to the surface in the channels of the tubular membranes, so that retained particles and larger molecules do not accumulate on the membrane surface. In this way, no layer of the filtered particles remains on the membranes [32]. The MF unit has an inlet for the wastewater and outlets for the retentate and permeate (filtrate) effluents. A high overflow rate (OVR) of 2–4 m/s, which is a measure of the velocity of the fluid inside the channels, allows for more effective removal of the retained particles due to turbulent flow and the concomitant large buoyancy forces. However, large OVR in combination with the existence of particles may lead to attrition of the thin microfiltration layer. For that reason, we worked with a total feed flow of 2–3 $m^3$/h, which gave a medium OVR of 0.13 m/s, and we applied frequent backpulsing (every 5 min) to remove the small cover layer that was formed. With the prerequisite that we do not have irreversible pore blocking, the momentary application of backflow, where the direction of solution's flow is reversed, is sufficient to restore the membrane's permeability. In this particular pilot system, backflow is applied with automatic timing, with the aim of preserving the membranes' good functioning and balancing their internal pressure.

The dimensions of both the NF unit outer shell and the components inside are estimated according to the requirements of productivity and purity of the water treated by the system. Additionally, optimal parameters of operation were determined. This included the following: (i) the dimensions of the NF unit to reach the required productivity; (ii) the materials and the thickness of the stainless steel shell and flanges to withstand the required operating pressure; (iii) the determination of the optimal influent pretreatment process prior the usage of NF, with the aim of ensuring continuous operation and minimizing the necessity for frequent membrane cleaning, while also mitigating the risk of irreversible membrane damage; (iv) design, engineering, study, development, and operation of a small prototype, including a single set of reactor internals, to test all of the components that will be used to seal the membranes, glass tubes, and glass sleeves, and the hermetical separation of the filtrate from the retentate at the bottom of the reactor unit; (v) design of the integrated treatment system laboratory-scale wastewater treatment; and (vi) design of the integrated wastewater treatment system at a 50 L/h aqueous fraction purification volume scale.

*Evaluation of Data*

Solid and liquid digestate samples were taken from the Lagada Biogas plant's anaerobic digester from each stage described above. The physicochemical and microbiological properties were measured and are presented in Tables 2–5 below.

General observations derived from Tables 2–4 and Figure 4 revealed that the solid digestate exhibits a higher abundance of nutrients, minerals, and macronutrients in comparison to all liquid digestates, indicating the efficacy of the processes of mechanical separation and centrifugation. It is also worth mentioning that the type of separation affected the accumulation of macromolecules and metals or micronutrients. In particular, the mechanical separation is less capable of removing total K and $NO_3^-$ as they appeared higher in content in $LD_{sep}$ 1.04 g/kg and 9.81 mg/kg compared to $SFD_{sep}$ 0.43 g/kg and 7.82 mg/kg, respectively. These findings are in agreement with those of a study by Popovic et al., who found that the separation efficiency can be enhanced by the addition of flocculants [33]. In the case of macronutrients Ca, Mg, S, and total N, mechanical separation with sieves ($SFD_{sep}$) was more effective for their transfer to the solid digestate in contrast to centrifugal separation ($SFD_{dec}$), which showed a better efficacy for Na, total P, and total K. Concerning trace elements and heavy metals, Hg, Cu, and B were better transferred to solid digestate using mechanical separation contrary to centrifugation, but the application of chitosan could increase centrifugation efficiency according to the results of Popovic et al. [33]. In the case of Cd, Cr, Ni, Zn, As, Fe, Mn, Si, and Al, a reverse phenomenon was observed as they were more efficiently transferred to solid digestate using centrifugation. It is observed that micronutrients can be adeptly transferred to the $SFD_{dec}$, which could be further applied as a solid organic soil improver.

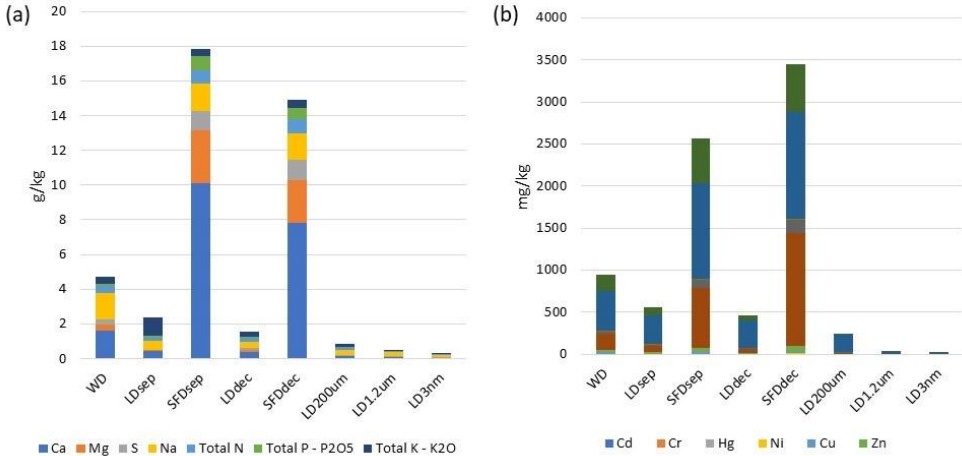

**Figure 4.** Process parameter variation. (**a**) Nutrient composition; (**b**) metal composition.

**Table 2.** Physicochemical characterization of retrieved fractions in each stage of treatment of the developed sanitation pilot unit.

| Parameter | WD | | Mechanical Separation | | | | Centrifugation | | | | Filtration | | uFiltration | | NF | |
| | WD ($n = 3$) | | $LD_{sep}$ ($n = 3$) | | $SFD_{sep}$ ($n = 3$) | | $LD_{dec}$ ($n = 3$) | | $SFD_{dec}$ ($n = 3$) | | $LD_{200\mu m}$ ($n = 3$) | | $LD_{1.2\mu m}$ ($n = 3$) | | $LD_{3nm}$ ($n = 3$) | |
| (unit) | Average | SD | Average | SD | Average | SD | Average | SD | Average | SD | Average | SD | Average | SD | Average | SD |
|---|---|---|---|---|---|---|---|---|---|---|---|---|---|---|---|---|
| pH | 7.8 | 0.09 | 8.2 | 0.10 | 8.6 | 0.11 | 8.3 | 0.10 | 8.5 | 0.10 | 8.3 | 0.10 | 8.3 | 0.10 | 8.3 | 0.00 |
| EC 25 °C (uS/cm) | 24,200 | 729 | 2830 | 311 | 2450 | 625 | 1874 | 150 | 1653 | 635 | 1400 | 49 | 340 | 32 | 281 | 14 |
| TS (%) | 5.23 | 0.20 | 2.66 | 0.10 | 29.03 | 1.12 | 1.91 | 0.07 | 20.41 | 0.79 | 0.77 | 0.02 | 0.23 | 0.01 | 0.16 | 0.01 |
| VS (%) | 3.82 | 0.00 | 1.61 | 0.00 | 23.97 | 0.00 | 1.29 | 0.00 | 13.96 | 0.00 | 0.59 | 0.00 | <0.05 | n.a. | <0.05 | n.a. |
| TSS (mg/kg) | - | - | 8226 | 1192 | - | - | 6133 | 889 | - | - | 4760 | 690 | 15 | 2.2 | 5.50 | 2.50 |
| VSS (mg/kg) | - | - | 984.5 | n.a. | - | - | 51 | n.a. | - | - | 33 | n.a. | 11 | n.a. | 6.00 | 0.00 |
| COD (mg $O_2$/kg) | 21,800 | 2108 | 18,400 | 1779 | 35,840 | n.a. | 16,200 | 1567 | 33,900 | 3278 | 7100 | 687 | 522 | 50.5 | 242 | 41.4 |
| Turbidity (NTU) | 40,365 | n.a. * | 12,372 | n.a. | n.a. | n.a. | 7646 | n.a. | n.a. | n.a. | 953 | n.a. | 264 | n.a. | 15.3 | 3.58 |
| $NO_3^-$ (mg/kg) | 12.3 | 2.09 | 9.81 | n.a. | 7.82 | 1.33 | 8.64 | 1.47 | 12.1 | 2.06 | 3.1 | 0.53 | 0.522 | 0.09 | 1.49 | 0.66 |

\* Not applicable.

**Table 3.** Macronutrient content of retrieved fractions in each stage of treatment of the developed sanitation pilot unit.

| Parameter | WD | | Mechanical Separation | | | | Centrifugation | | | | Filtration | | uFiltration | | NF | |
| | WD ($n = 3$) | | $LD_{sep}$ ($n = 3$) | | $SFD_{sep}$ ($n = 3$) | | $LD_{dec}$ ($n = 3$) | | $SFD_{dec}$ ($n = 3$) | | $LD_{200\mu m}$ ($n = 3$) | | $LD_{1.2\mu m}$ ($n = 3$) | | $LD_{3nm}$ ($n = 3$) | |
| (unit) | Average | SD | Average | SD | Average | SD | Average | SD | Average | SD | Average | SD | Average | SD | Average | SD |
|---|---|---|---|---|---|---|---|---|---|---|---|---|---|---|---|---|
| Ca (g/kg) | 1.63 | 0.14 | 0.43 | 0.04 | 10.11 | 0.87 | 0.39 | 0.03 | 7.81 | 0.67 | 0.12 | 0.01 | 0.06 | 0.01 | 0.03 | 0.01 |
| Mg (g/kg) | 0.33 | 0.03 | 0.09 | 0.01 | 3.01 | 0.27 | 0.12 | 0.01 | 2.49 | 0.22 | 0.04 | 0.00 | 0.03 | 0.00 | 0.02 | 0.00 |
| S (g/kg) | 0.32 | n.a. * | <0.05 | n.a. | 1.14 | n.a. | 0.12 | n.a. | 1.13 | n.a. | 0.07 | n.a. | 0.03 | n.a. | 0.02 | 0.00 |
| Na (g/kg) | 1.5 | 0.18 | 0.51 | 0.06 | 1.57 | 0.19 | 0.32 | 0.04 | 1.53 | 0.18 | 0.29 | 0.04 | 0.27 | 0.03 | 0.14 | 0.08 |
| $C_{org}$ (%) | 0.48 | 0.06 | 0.93 | 0.12 | 13.93 | 1.84 | 0.61 | 0.08 | 8.11 | 1.07 | 0.26 | 0.03 | <0.0003 | n.a. | <0.0003 | n.a. |
| Total N (g/kg) | 0.43 | 0.01 | 0.20 | 0.00 | 0.75 | 0.01 | 0.28 | 0.01 | 0.82 | 0.02 | 0.13 | 0.00 | 0.03 | 0.00 | 0.02 | 0.00 |
| $P_2O_5$ (g/kg) | 0.09 | 0.00 | 0.11 | 0.00 | 0.83 | 0.03 | 0.03 | 0.00 | 0.66 | 0.03 | 0.02 | 0.00 | 0.001 | 0.00 | 0.00 | 0.00 |
| $K_2O$ (g/kg) | 0.42 | 0.04 | 1.04 | 0.10 | 0.43 | 0.04 | 0.29 | 0.03 | 0.44 | 0.04 | 0.17 | 0.02 | 0.05 | 0.00 | 0.03 | 0.00 |

\* Not applicable.

**Table 4.** Micronutrient and heavy metal content of retrieved fractions in each stage of treatment of the developed sanitation pilot unit.

| Parameter | WD | | Mechanical Separation | | | | Centrifugation | | | | Filtration | | uFiltration | | NF | |
|---|---|---|---|---|---|---|---|---|---|---|---|---|---|---|---|---|
| | WD ($n=3$) | | $LD_{sep}$ ($n=3$) | | $SFD_{sep}$ ($n=3$) | | $LD_{dec}$ ($n=3$) | | $SFD_{dec}$ ($n=3$) | | $LD_{200\mu m}$ ($n=3$) | | $LD_{1.2\mu m}$ ($n=3$) | | $LD_{3nm}$ ($n=3$) | |
| (unit) | Average | SD | Average | SD | Average | SD | Average | SD | Average | SD | Average | SD | Average | SD | Average | SD |
| Cd (mg/kg) | 0.031 | 0.00 | 0.016 | 0.00 | 0.031 | 0.00 | 0.006 | 0.00 | 0.041 | 0.00 | <0.001 | n.a. * | <0.001 | n.a. | <0.001 | n.a. |
| Cr (mg/kg) | 0.55 | 0.09 | 0.39 | 0.06 | 1.91 | 0.31 | 0.21 | 0.03 | 4.14 | 0.68 | 0.13 | 0.02 | <0.02 | n.a. | <0.02 | n.a. |
| Hg (mg/kg) | 0.025 | 0.00 | 0.017 | 0.00 | 0.074 | 0.01 | 0.007 | 0.00 | 0.017 | 0.00 | <0.001 | n.a. | <0.001 | n.a. | <0.001 | n.a. |
| Pb (mg/kg) | 1.09 | 0.16 | 0.096 | 0.014 | 0.67 | 0.097 | 0.07 | 0.01 | 0.57 | 0.083 | <0.010 | n.a. | <0.003 | n.a. | <0.001 | n.a. |
| Ni (mg/kg) | 1.01 | 0.14 | 0.62 | 0.09 | 3.17 | 0.44 | 0.44 | 0.06 | 5.81 | 0.81 | 0.21 | 0.03 | 0.073 | 0.01 | 0.14 | 0.03 |
| Cu (mg/kg) | 20.8 | 1.90 | 1.83 | 0.17 | 17 | 1.55 | 2.74 | 0.25 | 16.7 | 1.52 | 0.9 | 0.08 | 0.06 | 0.01 | 0.05 | 0.01 |
| Zn (mg/kg) | 32.1 | 8.66 | 18.3 | 4.94 | 51.2 | 13.82 | 12.3 | 3.32 | 77.2 | 20.84 | 3.6 | 0.97 | 0.49 | 0.13 | 0.19 | 0.00 |
| As (mg/kg) | 0.29 | 0.04 | 0.12 | 0.02 | 0.23 | 0.03 | 0.04 | 0.01 | 0.23 | 0.03 | <0.005 | n.a. | <0.005 | n.a. | <0.005 | n.a. |
| Fe (mg/kg) | 189 | 45.98 | 78.5 | 19.10 | 713 | 173.47 | 46.9 | 11.41 | 1331 | 323.83 | 13.8 | 3.36 | 1.79 | 0.44 | 0.27 | 0.09 |
| Mn (mg/kg) | 24.8 | 2.38 | 14.4 | 1.38 | 91.9 | 8.81 | 7.3 | 0.70 | 167 | 16.02 | 4.6 | 0.44 | 0.066 | 0.01 | <0.025 | n.a. |
| B (mg/kg) | 5.56 | 0.99 | 3.59 | 0.64 | 15.9 | 2.84 | 3.61 | 0.64 | 13.3 | 2.38 | 1.85 | 0.33 | 1.17 | 0.21 | 1.54 | 0.16 |
| Si (mg/kg) | 475 | n.a. | 349 | n.a. | 1139 | n.a. | 327 | n.a. | 1260 | n.a. | 203 | n.a. | 20 | n.a. | 12.54 | 3.68 |
| Al (mg/kg) | 199 | 43.48 | 86.4 | 18.88 | 538 | 117.55 | 54.6 | 11.93 | 573 | 275.31 | 6.55 | 1.43 | 0.36 | 0.08 | 0.14 | 0.07 |

* Not applicable.

**Table 5.** Detection and quantification of indicator pathogen of retrieved fractions in each stage of treatment of the developed sanitation pilot unit.

| Parameter | WD | | Mechanical Separation | | | | Centrifugation | | | | Filtration | | uFiltration | | NF | |
|---|---|---|---|---|---|---|---|---|---|---|---|---|---|---|---|---|
| | WD ($n=3$) | | $LD_{sep}$ ($n=3$) | | $SFD_{sep}$ ($n=3$) | | $LD_{dec}$ ($n=3$) | | $SFD_{dec}$ ($n=3$) | | $LD_{200m}$ ($n=3$) | | $LD_{1.2\mu m}$ ($n=3$) | | $LD_{3nm}$ ($n=3$) | |
| (unit) | Average | SD | Average | SD | Average | SD | Average | SD | Average | SD | Average | SD | Average | SD | Average | SD |
| *Salmonella* spp. | N.D. * | n.a. ** | N.D. | n.a. | N.D. | n.a. | N.D. | n.a. | N.D. | n.a. | N.D. | n.a. | N.D. | n.a. | N.D. | n.a. |
| *Escherichia coli* (cfu/g) | 480 | 15 | 160 | n.a. | 310 | 11 | 120 | n.a. | 350 | 20 | <40 | n.a. | <9.1 | n.a. | <9.1 | n.a. |
| *Enterococcus faecalis* (cfu/g) | 830 | 29 | 650 | 31 | 770 | 25 | 390 | 13 | 620 | 19 | <40 | n.a. | <40 | n.a. | <9.1 | n.a. |

* Not Detected; ** not applicable.

As anticipated, the solids present in the liquid digestate are approximately 85% lower than those observed in the SFD. The microbial population is more pronounced in the whole digestate compared to the liquid fraction, as bacteria adhere to the solids, forming aggregates that provide protection against thermal and chemical alterations. However, it should be noted that the whole digestate exhibits significantly higher contamination levels than the solid digestate, indicating that the separation step contributes to microbial reduction. Regarding the microbiological characteristics of the digestate, it has been observed that potentially pathogenic aerobic microorganisms were present. Throughout the process, all pathogens demonstrated a reduction of 3–4 logarithmic units without the addition of spiked material. Moreover, the pathogenic microorganisms, *Escherichia coli* and *Enterococcus faecalis*, were better fractionated using centrifugation ($LD_{dec}$ = 120 cfu/g, $SFD_{dec}$ = 350 cfu/g and $LD_{dec}$ = 390 cfu/g, and $SFD_{dec}$ = 620 cfu/g, respectively) compared to mechanical separation ($LD_{sep}$ = 160 cfu/g, $SFD_{sep}$ = 310 cfu/g and $LD_{sep}$ = 650 cfu/g, and $SFD_{sep}$ = 770 cfu/g, respectively). It should be noted that *Enterococcus faecalis* is present in higher levels in the liquid phase in comparison to *Escherichia coli*, which is also more efficiently transferred to the solid phase. These findings are in agreement with a previous work of Díaz et al. according to which after centrifugation, Proteobacteria (*Escherichia coli*) move preferentially to the solid phase, contrary to Firmicutes (*Enterococcus faecalis*), which were the dominant phylum in the liquid phase [34]. Though there is a significant reduction in the microbial community, further biological treatment of the aforementioned liquid digestate is needed, to be suitable for reuse (Table 1).

The pH values remain consistent across all treatment processes, averaging at 8.3, indicating a stable pH level throughout all filtration and separation procedures (Figure 5). The conductivity values exhibit a significant decrease as the treatment processes progress from $LD_{dec}$ = 1874 μS/cm to $LD_{NF}$ = 281 μS/cm. This suggests the effective reduction of influent conductivity by the treatment processes. The percentage of TS gradually decreases from $LD_{dec}$ = 1.91% to $LD_{NF}$ = 0.16%, indicating successful removal of solid particles during the treatment processes. The VS percentage remains consistently low and falls below the detectable limit (<0.05%), indicating efficient removal of volatile solids during the treatment processes. COD values exhibit a decreasing trend from $LD_{dec}$ = 16,200 mg $O_2$/kg to $LD_{NF}$ = 242 mg $O_2$/kg 8 min, demonstrating the effectiveness of the treatment processes in reducing organic pollutants. Turbidity values decrease as the treatment processes advance, indicating the removal of suspended particles and improved water clarity. Notably, TSS values demonstrate a significant decrease from $LD_{dec}$ = 6133 mg/kg to $LD_{NF}$ = 5.50 mg/kg, indicating effective removal of suspended solids during the treatment processes. Similarly, the VSS values remain consistently low and fall below the detectable limit. *Salmonella* spp., *Escherichia coli*, and *Enterococcus faecalis* show values below the detection limit (<9.1 or N.D.), indicating successful microbial removal during the process.

The concentrations of elements, such as Ca, Mg, S, Na, total N, total P—$P_2O_5$, total K—$K_2O$, $NO_3^-$, Cd, Cr, Hg, Ni, Cu, Zn, As, Fe, Mn, B, Si, and Al, exhibit varying levels across the different treatment processes. Total N content after filtration in $LD_{1.2\mu m}$ and $LD_{3nm}$ was measured at 0.03 g/kg and 0.02 g/kg, respectively. Concerning, total P content in $LD_{1.2\mu m}$ and $LD_{3nm}$, it was measured at 0.001 g/kg and 0.00 g/kg, respectively, while total K was 0.05 g/kg and 0.03 g/kg, respectively. According to the limits set by the Hellenic Joint Ministerial Decision 145116/02-126 02-2011, which are presented in Table 1, the aforementioned $LD_{1.2\mu m}$ and $LD_{3nm}$ can only be used for limited irrigation and disposable cooling water, as turbidity and suspended solid measurements are higher than those determined. To this end, further treatment could be applied. Concerning micronutrient and heavy metal content, a significant reduction in the $LD_{sep}$ and $LD_{dec}$ apart from Cu and B was observed. Further analysis and comparison are required to assess the effectiveness of the treatment processes in reducing or removing these elements.

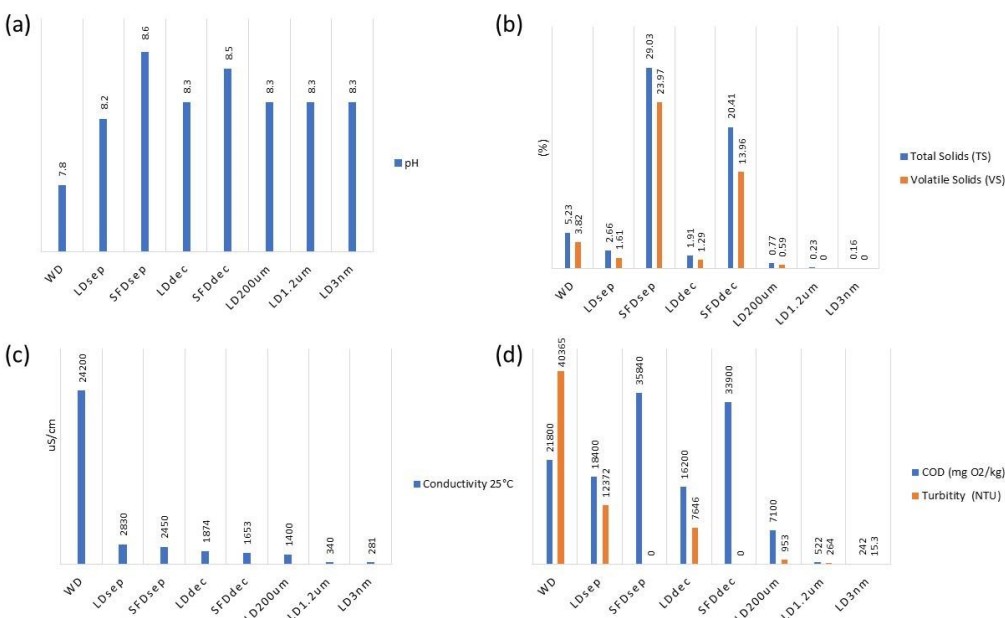

**Figure 5.** Process physicochemical parameter variation. (**a**) pH; (**b**) conductivity 25 °C; (**c**) total solids (TS) and volatile solids (VS); (**d**) chemical oxygen demand (COD) and turbidity.

Regarding the SFD$_{sep}$ and SFD$_{dec}$, the parameters presented in Tables 2–5 demonstrate that they represent concentrated fractions of the initial WD. The concentrations of various elements in the SFD$_{sep}$ and SFD$_{dec}$ fractions exhibit elevated levels in comparison to the WD. Similarly, the TS and COD values (Table 2) also display increased concentrations in the SFD$_{sep}$ and SFD$_{dec}$ fractions when compared to the WD. Indeed, the concentration was higher in the SFD$_{dec}$. The results highlighted that the mechanical separation and centrifugation could efficiently remove TS and COD from the liquid digestate at the first stages, because TS and COD were presented mostly in suspended particles rather than dissolved matter. For the particles with a size of <200 μm, the filtration of decreasing porosity that was used led to a TS value of LD$_{3nm}$ = 0.16% and COD value of LD$_{3nm}$ = 242 mg O$_2$/kg. It is worth mentioning that both values significantly decreased after 1.2μm filtration (Figure 5), and these findings are in agreement with those of a previous work by Akhiar et al. [35]. Graphical representations of these observations are given in Figure 5.

According to Regulation (EU) 2019/1009 and Table 6, a solid or liquid digestate obtained through anaerobic digestion may be used as EU fertilizing product or soil improver. In our case, SFD$_{sep}$ and SFD$_{dec}$ cannot be used as solid organic fertilizers as they are not in compliance with the macronutrient limits of total N, P$_2$O$_5$, and K$_2$O, but they can be utilized as soil organic improvers as they meet the European requirements. Furthermore, they are also in agreement with the limits set by China Organic Fertilizer Standard (NY525-2012) (COF Standards), California Code of Regulations (14CCR), and the Ohio Administrative Code for Land application system (Table 6).

It is known that the rules on land application of digestates may be laid down by each state in the United States. In addition, some states have laid down detailed guidelines or regulations for digestion; others may include digestates within the broader regulatory framework of waste disposal and nutrient management. Examining the case of California, all the retrieved fractions of Tables 3–5 during the different treatment stages can be utilized for land application.

**Table 6.** Requirements related to Fertilizing Products according to Regulation (EU) 2019/1009 of the European parliament, to California Code of Regulations (14 CCR) section 17852(a)(1), Ohio Rule 3745-42-13, and China's Organic Fertilizer Standards (NY 525-2012) (COF Standards) [11,36,37].

| | | Greek National Regulations | | US EPA 40 CFR Part 503 | Ohio Administrative Code | China Organic Fertilizer Standard (NY525-2012) (COF Standards) |
|---|---|---|---|---|---|---|
| | | European Regulation 2019/1009 | | California Code of Regulations (14 CCR) Section 17852(a)(1) | Rule 3745-42-13 \| Land Application Systems | NY/T 2596-2014 (CN) |
| | | Solid Organic Fertilizer (Shall Be in Solid Form) | Organic Soil Improver ($\geq$20% DM) | | | |
| Pollutants | Cd | $\leq$1.5 mg/kg DM | $\leq$2 mg/kg DM | $\leq$39 mg/kg DM | $\leq$0.01 mg/L | $\leq$3 mg/kg DM |
| | Cr (VI) | $\leq$2 mg/kg DM | $\leq$2 mg/kg DM | n.a. * | $\leq$0.1 mg/L | $\leq$150 mg/kg DM |
| | Hg | $\leq$1 mg/kg DM | $\leq$1 mg/kg DM | $\leq$17 mg/kg DM | $\leq$2 mg/L | $\leq$2 mg/kg DM |
| | Ni | $\leq$50 mg/kg DM | $\leq$50 mg/kg DM | $\leq$420 mg/kg DM | $\leq$0.2 mg/L | n.a. |
| | Pb | $\leq$120 mg/kg DM | $\leq$120 mg/kg DM | $\leq$300 mg/kg DM | $\leq$1.5 mg/L | $\leq$50 mg/kg DM |
| | As | $\leq$40 mg/kg DM | $\leq$40 mg/kg DM | $\leq$41 mg/kg DM | $\leq$0.1 mg/L | n.a. |
| | $C_2H_5N_3O_2$ | 0 | - | n.a. | n.a. | n.a. |
| | Cu | $\leq$300 mg/kg DM | $\leq$300 mg/kg DM | $\leq$1500 mg/kg DM | $\leq$0.2 mg/L | $\leq$100 mg/kg DM |
| | Zn | $\leq$800 mg/kg DM | $\leq$800 mg/kg DM | $\leq$2800 mg/kg DM | $\leq$2 mg/L | n.a. |
| Nutrients | N | >1% | n.a. | n.a. | $\leq$10 mg/L | n.a. |
| | $P_2O_5$ | >1% | n.a. | n.a. | n.a. | n.a. |
| | $K_2O$ | >1% | n.a. | n.a. | n.a. | n.a. |
| | $C_{org}$ | >15% | >7.5% | n.a. | n.a. | n.a. |
| Pathogen Indicators | *Salmonella* | 0 (n = 5) in 25 g or 25 mL | 0 (n = 5) in 25 g or 25 mL | $\leq$3 MPN/GTS | n.a. | n.a. |
| | *Escherichia coli* | $\leq$1000 (n,c = 5) g or mL | $\leq$1000 (n,c = 5) g or mL | n.a. | n.a. | n.a. |
| | *Enterococcaceae* | $\leq$1000 (n,c = 5) g or mL | $\leq$1000 (n,c = 5) g or mL | n.a. | n.a. | n.a. |
| | Fecal coliform | n.a. | n.a. | $\leq$1000 MPN/GTS | n.a. | n.a. |
| | BOD | n.a. | n.a. | n.a. | 40 mg/L | n.a. |
| Application Frequency | Land not zoned for agricultural uses | n.a. | n.a. | 1 time annually | 2 times annually | n.a. |
| | Land zoned | n.a. | n.a. | 3 times annually | 2 times annually | n.a. |
| Application Depth | | n.a. | n.a. | >12 inches accumulated on surface | >50 inches accumulated on surface | n.a. |

DM: dry matter; n: number of samples to be tested; c: number of samples where the number of bacteria expressed in CFU is between m and M; m: threshold value for the number of bacteria expressed in CFU that is considered satisfactory; M = maximum value of the number of bacteria expressed in CFU; MPN/GTS = most probable number per gram(s) of total solids; * Not applicable.

## 4. Conclusions

This study reported promising outcomes in purifying liquid digestate for several purposes such as irrigation and industrial use, while considering the design and engineering aspects. Treatment processes of the four described stages, including centrifugation, microfiltration, and nanofiltration, effectively reduced organic pollutants, eliminated suspended solids, and eliminated potentially pathogenic microorganisms. After the last NF stage, physicochemical analysis revealed stable pH levels, reduced conductivity to 281 $\mu$S/cm, and TS to 0.16% and VS to <0.05%. Furthermore, macronutrient, micronutrient, and heavy metal contents were significantly reduced, while enumeration of pathogenic microorganisms *Escherichia coli* and *Enterococcus faecalis* in $LD_{3nm}$ was <9.1 cfu/g. Moreover, improved water clarity, 15.3 NTU, was achieved through the treatment processes. Further analysis is necessary to evaluate the efficacy of the treatment process in reducing specific elements and ensuring compliance with regulatory standards. Additionally, SFDsep and SFDdec can be utilized as soil organic improvers as they meet the European requirements of Regulation (EU) 2019/1009.

These findings emphasize the potential of a sustainable alternative disinfection method and innovative technologies such as nanofiltration reactors in managing organic waste. Such approaches reduce pollution, safeguard agricultural practices, and promote public health and safety. Using treated and sanitized liquid digestate residue can boost sustainability on many fronts. Treated liquid digestate can be utilized for irrigation, disposable cooling water, and soil conditioners or organic liquid fertilizers, while the recovery of nutrients and water can be achieved, leading to low carbon emissions and a circular economy. Future studies should concentrate on improving existing methods and developing new procedures to help overcome the current difficulties associated with the high diversity of the matrix.

**Author Contributions:** Conceptualization, T.S.; methodology, T.S. and V.T.; software, T.S., A.G.C. and G.E.R.; validation, T.S. and G.E.R.; formal analysis, T.S. and G.S.; investigation, T.S., A.G.C., G.S. and V.T.; resources, T.S.; data curation, T.S. and A.G.C.; writing—original draft preparation, G.S., T.S. and A.G.C.; writing—review and editing, T.S., G.S., G.E.R. and P.F.; visualization, T.S., A.G.C. and G.S.; supervision, T.S.; project administration, T.S.; funding acquisition, T.S. All authors have read and agreed to the published version of the manuscript.

**Funding:** This research was co-funded by the European Regional Development Fund of the European Union and Greek national funds through the Operational Program Competitiveness, Entrepreneurship, and Innovation, under the call RESEARCH—CREATE—INNOVATE (project code: T2EDK-04043).

**Institutional Review Board Statement:** Not applicable.

**Informed Consent Statement:** Not applicable.

**Data Availability Statement:** The data presented in this study are available, upon request, from the corresponding author.

**Acknowledgments:** The authors wish to acknowledge operational manager Stefanos Patsatzis of Biogas Lagada S.A. for the industrial tests and the provision of appropriate quantities of digestate. The authors also wish to acknowledge all staff members of Qlab P.C. for their individual roles that contributed to the implementation of this study, thank you Vasiliki Tsioni, Ioanna Dalla, Ifigeneia Grigoriadou, Ioanna Christoforidou, and Panagiotis Pantazis.

**Conflicts of Interest:** The authors declare no conflict of interest.

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
