# Peer review of "Fractionation of Anaerobic Digestion Liquid Effluents through Mechanical Treatment and Filtration"

_sustainability, doi:10.3390/su151411178_

Round 1
Reviewer 1 Report
1. Section 1. Line 63-101. Authors described the related works in a detail manner in the respective paragraphs. In my opinion, the information given was way too much. Instead, authors should highlight the important key points, followed by the relation to the current work. Laboratory scale works were not mentioned and would be good to include some (Camilleri-Rumbau et al., 2021; Piccoli et al., 2021; Zheng et al., 2016). Please also address the knowledge gaps based on the related works mentioned, then link to the novelty of this work.
2. Section 1. Line 103-105. The digestate used as fertilizer regulation should be given a reference. Apart from EU regulation, we should consider other countries’ regulation as well, such as US and China regulation? Also, please mention the set limit for the respective countries’ regulation, a table will be good to have a better comparison purpose, similar like Table 1.
3. Section 2.1. Please include a figure of the WD treatment equipment system, and also the biogas plant. A figure of the overall integration of both systems would be good.
4. Section 2.3. In my opinion, the methodology for the respective physicochemical methods were way too detailed. As these methods are standardized, which can be found easily from the manufacturer’s protocol online, my suggestion is to simplify them accordingly, by mentioning the key methods, then cite the manufacturer’s protocol will do. For instance, “the TS and VS determination were according to the Standard Methods for the Examination of Water and Wastewater (Clesceri et al., 1998)”.
5. Section 3.1. Please revise the table numbering. Please include statistical analysis for the t-test, and also the number of replicates performed.
6. Line 394-396. In my opinion, the N and P tends to increase in the liquid fraction, that could act as mineral fertilizers (nitrogen fertilizers) due to its high ammonia content (Tambone et al., 2017), which was a different scenario based on this work. Why is it so?
7. Line 403-412. Microbial population diversity was not discussed? What are the significant bacterial and archaea phyla/ genera, as well as the microbial difference between solid and liquid digestate?
8. Line 413. pH 8.3 is considered high pH for digestion. Does this pH refer to the digestate before nitrification? pH of the digestate should be decreased after the separation (Pelayo Lind et al., 2021).
9. Line 432-433. Elemental composition was mentioned without any further discussion. Please clarify and compare with the regulations mentioned in the Introduction section.
10. Line 437-438. Please give a reason for the decrease for both TS and COD values after separation with sieves/ centrifugal separation? How about COD removal efficiency? COD removal efficiency is important as to determine the efficiency of the biodegradable organic matter rate with the separated digestate liquid.
11. Overall aspects of the results and discussions section. I personally feel there’s a lack of discussion information, no detailed supporting documentation to support the results obtained, not to say the results comparison with other works. A major revision of this section is deemed necessary.
12. Conclusion should be short and precise. Straight to the key points/ results and how it could help/ benefits to the society and economic development.
13. There are several grammatical errors throughout the manuscript. English language needs to be further revised.
References
Camilleri-Rumbau, M.S., Briceño, K., Søtoft, L.F., Christensen, K.V., Roda-Serrat, M.C., Errico, M., Norddahl, B., 2021. Treatment of manure and digestate liquid fractions using membranes: Opportunities and challenges. International Journal of Environmental Research and Public Health 18, 1–30. https://doi.org/10.3390/ijerph18063107
Clesceri, L.S., Greenberg, A.E., Eaton, A.D., Eds., 1998. Standard Methods for the Examination of Water and Wastewater, 20th ed. APHA, AWWA, WEF: Washington, DC.
Pelayo Lind, O., Hultberg, M., Bergstrand, K.J., Larsson-Jönsson, H., Caspersen, S., Asp, H., 2021. Biogas Digestate in Vegetable Hydroponic Production: pH Dynamics and pH Management by Controlled Nitrification. Waste and Biomass Valorization 12, 123–133. https://doi.org/10.1007/s12649-020-00965-y
Piccoli, I., Virga, G., Maucieri, C., Borin, M., 2021. Digestate liquid fraction treatment with filters filled with recovery materials. Water (Switzerland) 13, 1–12. https://doi.org/10.3390/w13010021
Tambone, F., Orzi, V., Imporzano, G.D., Adani, F., 2017. Solid and liquid fractionation of digestate : Mass balance , chemical characterization , and agronomic and environmental value. Bioresource Technology 243, 1251–1256.
Zheng, Yangqing, Ke, L., Xia, D., Zheng, Yanmei, Wang, Y., Li, H., Li, Q., 2016. Enhancement of digestates dewaterability by CTAB combined with CFA pretreatment. Separation and Purification Technology 163, 282–289. https://doi.org/10.1016/j.seppur.2016.01.052
There are a number of grammatical errors, and the entire manuscript can be written better.
Reviewer 2 Report
The study is about treating the liquid digestate from anaerobic digestion. Overall the study is highly represented what the authors would like to investigate. I recommend the folling works to read and cite: (2020) Impact of mixing intensity and duration on biogas production in an anaerobic digester: a review, Critical Reviews in Biotechnology, 40:4, 508-521, DOI: 10.1080/07388551.2020.1731413
In that study the authors investigate the effect of mechical treatment for the microbacterial communities. That is a very crutial point if the microbiological communities would be utilized for further purposes. In your article should be mention the mechanical treatment's effect on the microbiology. What you will do with the bacterias after the treatment?
I propose to consider my suggestion.
Reviewer 3 Report
In this study, anaerobic digestion liquid digestate is separated by mechanical treatment and filtration to achieve effective management and low hazard utilization of liquid digestate. But the study needs significant improvement before acceptance for publication. I recommend rejection for this manuscript with detailed comments as follows:
1. Lines 56-57, Please carefully check the format to avoid unnecessary line breaks.
2. The description of other references in the Introduction should be reduced and the Introduction should be focused on the gap to be addressed in this study.
3. The serial number of the tables and figures need to be rechecked to ensure that it corresponds to the text section in the manuscript. In addition, both the figures and tables need to be displayed in the main text to better understand the research content and results.
4. There are too many paragraphs in this manuscript. Suggest merging the related content into one paragraph.
5. There is almost no discussion in the manuscript, just a description of the data. It is recommended to add a lot of discussion.
6. The figures and tables need to be further beautified to achieve better results.
Extensive editing of English language required
Reviewer 4 Report
Manuscript title: Fractionation of anaerobic digestion liquid effluents through mechanical treatment and filtration
This paper is well written and the methodology tested by the authors for digestate treatment of filtration (solid-liquid separation, microfiltration and nanofiltration) after anaerobic digestion is highly significant addition to the energy production technology. The basic idea, hypothesis, introduction and methodology is well presented.
I would suggest including discussion and comparing the results with previous studies.
The discussion need improvement
Round 2
Reviewer 1 Report
This revision is acceptable.
Author Response
We would like to thank the reviewer.
Reviewer 3 Report
In this study, anaerobic digestion liquid digestate is separated by mechanical treatment and filtration to achieve effective management and low hazard utilization of liquid digestate. But before this study can be accepted for publication, the following modifications need to be made.
1. Line 62, which sentences are these references cited after?
2. The order of the tables needs to be adjusted based on the order appearing in the main text. For example, Table 3 appears earlier than Table 2 in the manuscript.
3. Sentences with strikethrough lines should not appear in revised manuscripts.
4. Suggest changing the color of the text in Figure 3 and Figure 4 to black to express the data more clearly.
5. Section 2. The Materials and Methods section contains a lot of content, and it is recommended to reduce this section and describe it in a more precise way.
6. Line 521, Please revise the figure numbering and carefully check the number of the figure in the entire manuscript. There are two "Figure 3" in this manuscript.
Minor editing of English language required
